# Perceived Training of Junior Speed Skaters versus the Coach’s Intention: Does a Mismatch Relate to Perceived Stress and Recovery?

**DOI:** 10.3390/ijerph191811221

**Published:** 2022-09-07

**Authors:** Ruby T. A. Otter, Anna C. Bakker, Stephan van der Zwaard, Tynke Toering, Jos F. A. Goudsmit, Inge K. Stoter, Johan de Jong

**Affiliations:** 1School of Sports Studies, Hanze University of Applied Sciences, 9747 AS Groningen, The Netherlands; 2Section Anatomie & Medical Physiology, Department Biomedical Sciences of Cells & Systems, University Medical Center Groningen, University of Groningen, 9713 GZ Groningen, The Netherlands; 3Department Human Movement Sciences, University Medical Center Groningen, University of Groningen, 9713 GZ Groningen, The Netherlands; 4Department of Human Movement Sciences, Vrije Universiteit Amsterdam, 1081 HV Amsterdam, The Netherlands; 5Leiden Institute of Advanced Computer Science, Universiteit Leiden, 2300 RA Leiden, The Netherlands; 6School of Sport Studies, Fontys University of Applied Science, 5612 MA Eindhoven, The Netherlands; 7Department of Industrial Design, Eindhoven University of Technology, 5600 MB Eindhoven, The Netherlands; 8Innovatielab Thialf, 8443 DA Heerenveen, The Netherlands

**Keywords:** training load, adolescent athletes, perceived stress and recovery, monitoring, coaching

## Abstract

The aim of this observational study was to examine the differences between training variables as intended by coaches and perceived by junior speed skaters and to explore how these relate to changes in stress and recovery. During a 4-week preparatory period, intended and perceived training intensity (RPE) and duration (min) were monitored for 2 coaches and their 23 speed skaters, respectively. The training load was calculated by multiplying RPE by duration. Changes in perceived stress and recovery were measured using RESTQ-sport questionnaires before and after 4 weeks. Results included 438 intended training sessions and 378 executed sessions of 14 speed skaters. A moderately higher intended (52:37 h) versus perceived duration (45:16 h) was found, as skaters performed fewer training sessions than anticipated (four sessions). Perceived training load was lower than intended for speed skating sessions (−532 ± 545 AU) and strength sessions (−1276 ± 530 AU) due to lower RPE scores for skating (−0.6 ± 0.7) or shorter and fewer training sessions for strength (−04:13 ± 02:06 hh:mm). All training and RESTQ-sport parameters showed large inter-individual variations. Differences between intended–perceived training variables showed large positive correlations with changes in RESTQ-sport, i.e., for the subscale’s success (*r* = 0.568), physical recovery (*r* = 0.575), self-regulation (*r* = 0.598), and personal accomplishment (*r* = 0.589). To conclude, speed skaters that approach or exceed the coach’s intended training variables demonstrated an increased perception of success, physical recovery, self-regulation, and personal accomplishment.

## 1. Introduction

Speed skating is a competitive sport in which a multidisciplinary training program is required for the athletes to perform well at the elite level. Training programs for speed skaters include aerobic-, anaerobic-, and strength training on ice, on the bike, and in the gym [1]. This emphasizes the need for speed skating coaches to find a good balance in their training programs so that the training volume and intensity can be managed by the athletes. The Dutch Speed Skating Association selects speed skaters from the age of 17 years and provides them with a high-quality training program [2]. For many speed skaters, this is the time during which they rapidly increase their training volume to be ready for the senior level, which starts after the age of 19 years. The increasing training volume, in addition to the fact that junior athletes develop in many domains other than sport as well (e.g., school, friendships, etc.), may make each individual vulnerable to non-functional overreaching, overtraining [3], and overuse injuries [4]. On the other hand, too few training loads can lead to non-optimal performance increases. Nearly thirty percent of young athletes experience non-functional overreaching or overtraining at least once in their careers [5]. Therefore, it is important to prescribe a tailored training load for individual athletes.

Coaches plan training sessions based on their point of view of training intensity distribution, taking the frequency, intensity, and duration of sessions into account [6]. Usually, the intended training load (defined as the sum of all training sessions’ intensities multiplied by the duration [7]) is prescribed as an external load such as speed, pace, duration, or power. However, the internal training load response of the athlete determines the outcome of the training [8]. The internal load (psychophysiological response) is in turn dependent on the actual external load and personal circumstances of the athletes, such as age [9], experience [10], and perceived stressors [11].

A reliable, valid, and also easy way to monitor internal training load is by logging session Ratings of Perceived Exertion (sRPE) and duration for each training session [12]. This method has shown to be useful for a wide variety of training modalities, such as endurance, interval, and strength training [13]. It can be used for both the prescribed training sessions (intended load) by the coach and the perception of actual training sessions (perceived load) by the athlete so that a comparison can be made [14].

Many studies have shown that there are differences between the intended training load by the coach and the perceived training load by the athletes, using measurements of sRPE and duration. In an attempt to specify the differences, the training sessions have been divided into easy, moderate, and hard. On the RPE CR10 scale [12], easy, moderate, and hard were classified as an RPE < 3, 3–5, and >5, respectively [14,15,16,17]. However, studies using the 6 to 20 Borg scale [18] have defined two different classifications for easy, moderate, and hard, i.e., RPE < 11, 11–14, and >14, respectively [19], and RPE < 13, 13–14, and >14, respectively [20]. These differences in classifications lead to differences in interpretation of the results, which makes it hard to compare different studies. In order to have a rough overview of previous studies, a short summary of the results in cyclic sports is given.

Previous studies using the RPE CR10 scale showed no differences in intensity, duration, and training load on average [14,15,16,17]. However, the training load of easy sessions was perceived as higher by the athletes. Intended moderate sessions did not show any differences between coach and athlete, except for male cross-country runners who perceive it as harder [17]. Moreover, the training load of intended hard sessions was perceived as lower by runners [14], swimmers [16], and female cross-country runners [17]. For cyclists [15] and male cross-country runners [17], there were no differences for intended hard sessions. Using the Borg scale, it was shown that semi-professional cyclists perceived both intended easy sessions and hard sessions as easier [19].

Overall, it seems that there is quite some variation between the perception of sessions that are meant to be easy, moderate, and hard, while the total training duration and training load are mainly similar. One reason for the variation may be that both adolescent and adult athletes were included in the studies. It has been shown that, during adolescence, the correlation between RPE and heart rate [9], and between the coach’s and athlete’s RPE, increases with age [10]. This suggests that as adolescent athletes reach adulthood, they may vary less in their appraisal of exertion. Furthermore, there seems to be a difference in several sports modalities such as running, cycling, and swimming. Given that the division of the scales can have a great influence on the results, it is important to visualize the entire width of the scale that has been used to gain more detailed information.

Kenttä and Hassmén’s (1998) conceptual model shows that the accumulation of physical and psycho-social stress and/or a shortage of recovery may lead to maladaptations such as decreased performance and an increased injury risk [21]. A consensus statement reads that the outcome of training stimuli is, among other factors, influenced by burdening psycho-social factors [3]. In addition, a recent review described that increased perception of stress is one of the main indicators of functional overreaching [11]. Therefore, regular measurement of perceived stress and recovery can be useful for athletes to prevent maladaptation to training.

Intended training load by the coach and perceived training load by junior speed skaters, and what that means for perceived stress and recovery, have not been investigated. Therefore, the goal of this study is two-fold: (1) to examine the (mis)match between the intended training intensity, duration, and load by the coach and the perceived training intensity, duration, and load by junior speed skaters over the entire range of intensities and for the different modalities of training (skating, cycling, strength training, and other); and (2) to explore how the (mis)match in training variables relates to changes in the perception of stress and recovery by the speed skaters in a four-week preparatory period.

## 2. Materials and Methods

### 2.1. Subjects

In this study, the whole regional talent team including 23 junior speed skaters (14 males, 9 females; age, 18 ± 1 y (range: 16–19 y; body mass, 68 ± 7 kg; height, 178 ± 8 cm) and their two coaches participated. The speed skaters were part of one training group divided over two coaches. They all performed at a national and international level with an average personal best 1500 m time 12% above the WR. In speed skating, this equals the sub-elite level [22]. All speed skaters and coaches were informed about the procedures of the study and signed a written informed consent form. This study was approved by the ethics committee of the Department of Human Movement Sciences, University of Groningen, and was in accordance with the Declaration of Helsinki.

### 2.2. Design

In this observational study, the group of 23 junior speed skaters was monitored over a four-week period. The four weeks were during the general preparation period (June–August 2021), in which the speed skaters executed different training modalities, i.e., speed skating, cycling, strength training, and other training modalities (such as plyometric training and roller skating). For every training session, the intended training as planned by the coach for each individual athlete and the actual training as perceived by the athlete were monitored. All training modalities were included in this study.

### 2.3. Training Load

During the four weeks, the coaches provided each individual speed skater with a weekly training schedule, which was shared in advance. The schedule included the number of sessions, training modality, a description of the type of training, intensity, the duration of specific parts of the training, and the total intended duration. To define the intended training intensity for each training session, the coaches also filled in the intended Rating of Perceived Exertion (RPE) for each session in a separate spreadsheet. The intended RPE was not shared with the speed skaters. To determine the perceived training load by the speed skaters, the Smartabase athlete management platform was used (version 6.11.6). In this platform, speed skaters filled in the realized training modality, the duration, and the perceived RPE for each training session. The athletes were instructed to do this approximately 30 min after each training session [12]. For the RPE, a modified Borg CR10 scale from 0 (rest) to 10 (maximum) was used [12]. Before the four-week period, all speed skaters and coaches were familiarized with the modified Borg CR10 scale. Intended and perceived training load were calculated using the following formulas [12]:Coach’s intended training load (AU) = intended RPE × intended duration.
Athlete’s perceived training load (AU) = RPE × duration.

### 2.4. Recovery and Stress

Before and after the four weeks of training, RESTQ-sport questionnaires were administered to all speed skaters to define perceptions and sources of recovery and stress [23]. In addition, four weeks prior to the start of the study, a RESTQ-sport was administered to familiarize the speed skaters with the questionnaire. Detailed information about the RESTQ-sport can be found in Kellmann and Kallus [23]. In this study, the Dutch version of the RESTQ-sport was used, which has shown sufficient reliability and validity [24]. The RESTQ-sport was filled out within three days before the start and the end of the four-week monitoring period using the Smartabase athlete management platform (version 6.11.6).

### 2.5. Inclusion Criteria

Two criteria were used to determine if a speed skater could be included in the analyses. First, the speed skaters needed to complete both RESTQ-sport questionnaires. Secondly, to correct for not filling in the training data, speed skaters had to execute and fill in at least 60% of the intended training load to be included in the analysis. If there was reasonable doubt by one of the researchers in consultation with the coach whether a training log was filled in correctly, all data of that training session were excluded.

### 2.6. Statistical Analyses

Means and standard deviations of the intended and perceived number of sessions, training load, training load per session, RPE per session, duration, and duration per session were calculated for the four-week monitoring period, compiling all training modalities and for each training modality separately. For these variables, means and standard deviations were also calculated for the different scores of perceived minus intended training variables, which is the difference between coach and athlete. Furthermore, means of the RPE distribution over the four-week period were calculated for intended sessions by the coach and perceived sessions by the speed skaters. To determine differences between intended and perceived training variables, Mann–Whitney U tests were performed for all variables mentioned above. For the different training modalities, percentages of performed training durations were also calculated.

To show differences in the pre- and post-RESTQ-sport scores, means were calculated for the group of speed skaters. To test differences in the pre- and post-RESTQ-sport scores, a Wilcoxon signed-rank test was performed for the aggregated scores of general stress, general recovery, sport-specific stress, sport-specific recovery, total stress, total recovery, and for the recovery–stress balance. For these variables, a difference score between the pre- and post-RESTQ-sport scores was also calculated.

To test if there was a relationship between perceived stress and recovery and differences in training variables intended by the coach and perceived by the speed skaters, correlations between a difference score of the RESTQ-sport measurements and a difference score of training load, RPE per session, and duration per session were calculated. The difference score of the RESTQ-sport was defined as the post-measurement minus the pre-measurement. This was calculated for all variables and for the aggregated variables (i.e., general stress, general recovery, sport-specific stress, sport-specific recovery, and recovery–stress). The difference scores for training characteristics were defined as the perceived value minus the intended value. Before calculating the correlations, the scores were checked for normality. In the case of normality, Pearson correlations were calculated. Otherwise, Spearman correlations were calculated. Descriptive statistics, *t*-tests, and correlations were calculated using SPSS (version 27.0; SPSS, Inc., Chicago, IL, USA). Statistical significance was set at *p* < 0.05. The magnitude of the difference between the measures (effect size) was considered as <0.3 small, 0.3–0.5 moderate, and >0.5 large [25]. The magnitude of correlations are inferred as <0.1 trivial, 0.1–0.3 small, 0.3–0.5 moderate, 0.5–0.7 large, 0.7–0.9 very large, and 0.9–1.0 almost perfect [26].

## 3. Results

Of the initial 23 speed skaters, 8 did not reach 60% compliance with the intended training load, and 1 speed skater did not complete both RESTQ-sport questionnaires. Consequently, nine speed skaters were excluded for analysis. In total, data from 438 and 378 training sessions were collected from both coaches and speed skaters over four weeks, respectively. Of the remaining 14 speed skaters, 11 executed fewer training sessions than intended (range: 1 to 15 sessions less) and 3 speed skaters executed more sessions than intended (range: 1 to 6 sessions more). Table 1 shows an overview of the intended and perceived training characteristics over the four-week monitoring period.

Over the entire four weeks, the intended number of sessions was significantly higher compared to the perceived number of sessions with a moderate effect size (*U* = 51.50, *z* = −2.15, *p* = 0.031, *r* = −0.41), shown in Table 1. The intended training duration was also higher compared to the perceived duration and showed a moderate effect size (*U* = 52.00, *z* = −2.11, *p* = 0.035, *r* = −0.40). Other training variables did not differ over the four weeks, *p* > 0.05.

To provide insight into the differences between and the contributions of the training modalities, Table 2 shows an overview of the intended and perceived training characteristics for the different training modalities. For speed skating, a higher intended versus perceived total training load (*U* = 45.00; *z* = −2.44; *p* = 0.014; *r* = −0.46), RPE per session (*U* = 45.00; *z* = −2.45; *p* = 0.014; *r* = −0.46), and load per session (U = 48.50; z = −2.28; *p* = 0.021; *r* = −0.43) were found, showing moderate effect sizes. In addition, for strength training, a higher intended number of sessions (*U* = 10.50; *z* = −4.12; *p* < 0.001; *r* = −0.78), training load (*U* = 21.50; *z* = −3.53; *p* < 0.001; *r* = −0.67), training duration (*U* = 7.00, *z* = −4.26; *p* < 0.001; *r* = −0.80), and duration per session (*U* = 52.00; *z* = −2.34; *p* = 0.035; *r* = −0.44) were found compared to perceived variables along with moderate to large effect sizes. Furthermore, other training modalities showed a moderately higher intended duration per session (*U* = 44.50, *z* = −2.46, *p* = 0.014, *r* = −0.46) and a moderately higher load per session (*U* = 53.50; *z* = −2.05, *p* = 0.039, *r* = −0.39). No differences were found between intended and perceived training characteristics for cycling, *p* > 0.05. Of the total training duration over four weeks, the athletes performed 12.6 ± 3.4%, 52.6 ± 16.4%, 12.6 ± 3.5%, and 22.2 ± 15.6% of speed skating, cycling, strength training, and other training modalities, respectively.

The RPE distribution over the entire four weeks for all training modalities is shown in Figure 1. Significant differences between the number of intended and perceived sessions on a certain RPE were found on RPE 4, (*U* = 46.00; *z* = −2.40; *p* = 0.016; *r* = −0.45), RPE 6, (*U* = 12.00; *z* = −4.01; *p* < 0.001; *r* = −0.76), and RPE 8 (*U* = 53.50; *z* = −2.11; *p* = 0.03; *r* = −0.40), all with moderate to large effect sizes. In all cases, the intended number of sessions for that RPE was higher compared to the perceived number of sessions (Figure 1).

Figure 2 shows the RESTQ-sport values before and after the four-week monitoring period, and Table 3 shows the differences between the aggregated scores of both RESTQ-sport measurements. Both measurements look comparable upon observation, with lower scores on stress scales compared to higher scores on recovery scales (Figure 2). The only slight variations between pre- and post-RESTQ-sport scores can be viewed for emotional stress, sleep quality, and self-efficacy. This is confirmed by no differences between the pre- and post-RESTQ-sport scores on the aggregated variables (i.e., general stress, general recovery, sport-specific stress, sport-specific recovery, and recovery–stress; range difference: −0.8–0.3; *p* > 0.05).

Table 3 shows the correlations between individual changes in RESTQ-sport scores and the difference between intended and perceived training variables. Significant correlations were found between differences in total training load and personal accomplishment, between RPE per session and success, and between duration per session, physical recovery, and self-regulation, *p* < 0.05. All significant correlations were of a large magnitude (*r* > 0.56).

## 4. Discussion

The first aim of this observational study was to examine the (mis)match between the intended training intensity, duration, and load by the coach and the perceived training intensity, duration, and a load of junior speed skaters for training sessions at all intensities and for all training modalities. The second aim was to explore the relationship between the difference between intended and perceived training load and changes in perception of stress and recovery of the speed skaters in a four-week preparatory period.

In summary, the study results showed that the average load and RPE per session were similar for the coach and the speed skaters, though athletes performed, on average, four sessions less in the four-week training period, resulting in lower total training duration, but no difference in the total training load. However, differences between the coach and the individual speed skaters show a very large variation between individual speed skaters with a range of 15 sessions less to 6 sessions more than intended. The training intensity distribution of the sessions reported by the speed skaters was moderate to largely lower than intended by the coach for the RPE scores of 4, 6, and 8. Over the four-week preparatory period, there was no change in perceived stress and recovery on average for the group, but the variation in changes between speed skaters was rather large (SDs ranging from 0.39 to 1.07). The individual change in the RESTQ-sport subscales success, physical recovery, self-regulation, and personal accomplishment showed large positive correlations with the difference between intended and perceived training variables.

This is the first study to show the intended and perceived training intensity, duration, and load of talented junior speed skaters. It was shown that the training load of junior talented speed skaters is lower than what the coach has intended because speed skaters execute fewer sessions. There are no differences in RPE and duration between the coach and the athlete per session including all training modalities. This was also found in a study among elite cyclists [19]; however, a study of young soccer players showed that the athletes trained at a higher RPE than intended [20]. On the other hand, our study shows that the skater’s RPE for speed skating sessions is 0.6 lower than what the coach intended. Remarkably, to the best of our knowledge, no other study has shown a lower perceived RPE, on average, than intended by the coach. A reason for the difference may be, for example, because the speed skaters performed their sessions with a lower external load (speed) than intended or because the coach underestimated the physical capacity of the athletes. Additionally, the included skaters were in their late adolescence or early adulthood from an age of 16 to 19 years old. It has been shown that the reliability of the RPE scale improves from childhood to adulthood [9]. Additionally, another study showed that the correlation between the intended RPE of the coach and the perceived RPE of swimmers increases with age and experience [10]. This may explain the large variation of our results, as we included adolescents. Note that it is important that the instructions for how to use the RPE scales are clear to both the coach and speed skaters to align their RPE ratings. In addition, it has been shown that the reliability of RPE can be improved for adolescents when using the OMNI scale [9], and recent findings show that adding facial expressions to the RPE scale is more convenient to use and provides valid and reliable scores for training monitoring [27]. Further research should analyze the reasons why RPE scores from athletes could be lower than those intended by the coach for speed skating sessions and explore the use of facial expressions added to RPE.

Another interesting finding is that the intended training intensity distribution was in line with the training intensity distribution of an Olympic 1500 m medalist [28]. However, the perceived training intensity distribution in our study seems to differ. In 2006 and 2007, the Olympic 1500 m medalist showed a frequency distribution with a peak in the number of sessions at an RPE of four [28]. For the same athlete, a steeper peak was shown in 2008 and 2009 at an RPE of three. Our study shows that the intended training intensity distribution by the coach was similar to the profile of the Olympic medalist in 2006, with a peak at an RPE of four [28]. However, the perceived training intensity distribution is more flattened with a plateau at RPEs of three, four, and five of which the performed sessions at an RPE of four and eight were moderately lower than intended by the coach, and at an RPE of six, were largely lower than intended.

When putting our findings into a three-zone perspective (low, moderate, and high based on RPE < 3, 3–5 and >5, respectively) [14], our study shows that athletes spend less training sessions at moderate and hard intensity than intended by the coaches (see Figure 1). Other studies have shown a higher duration at low intensity, the same or higher duration at moderate intensity, and the same or lower duration at high intensity [14,15,16,17]. In our study, training duration was different due to the number of sessions, while other studies showed differences due to a different duration per session. These discrepancies indicate that it is crucial to look at the training duration, intensity, and load as a whole but also at all separate sessions of each individual athlete, coach, and group in order to find a suitable intervention to align the intention and the perception.

A limitation of this study is that only 14 speed skaters out of the initial 23 speed skaters met the inclusion criterium of executing >60% of the intended training load. This resulted in 438 intended sessions by the coach and 378 executed sessions by the skaters. This difference may be due to skaters that did not fill out the training log or did not execute the intended training sessions. In order to ensure as little bias as possible due to not filling out the training log, we have discussed each missing training session with the coach leading to the inclusion criterium of >60%, which ensured that the speed skaters who did not fill out the training log structurally were excluded. Therefore, we are confident that the sessions that were not filled out by the skaters have not been executed. Reasons for not executing these sessions could be a vacation, other obligations, injuries, or illness.

Some studies comparing intended and perceived training load did not report a discrepancy in the number of training sessions [14,15,20] or showed a very small discrepancy [17,19]. If no discrepancy was shown, it was due to the design, which would only include matched sessions. The advantage of including only matching sessions is that a good comparison for each session can be made. However, a clear overview of the total training load that was intended by the coach and the training load that was actually executed by the athletes could not be given, which makes it difficult to make inferences on under- or overtraining. We chose to include all sessions because that gives a realistic overview of the total training load in relation to perceived stress and recovery.

Studies that showed a very small discrepancy included athletes that probably reported their training load comprehensively because it is their profession [19], and because most training sessions were performed under the supervision of the coach [17]. During our study, the junior speed skaters had to do the majority of training sessions on their own initiative such as cycling and other sessions, which leaves more room for absence if they were, e.g., otherwise occupied, or simply not in the mood for training. This is probably the reason why we found differences in the number of sessions between the coach and the athlete. The fact that the coach was not present during most training sessions emphasizes the need for monitoring training load, stress, and recovery in order to give the coach a good overview.

The speed skaters in our study perceived greater success, physical recovery, self-regulation, and personal accomplishment when their training load was approaching or exceeding the intention of the coach. This was shown by a positive correlation between success and the difference between the intended–perceived RPE per session, physical recovery, and self-regulation with perceived–intended duration per session, and personal accomplishment with perceived–intended total training load. However, we did not find significant relationships between differences in perceived–intended training duration, RPE, load, and aggregated scores of total stress and total recovery.

Some studies show a relation between increased training load or a negative life event and success, physical recovery, and self-regulation [29,30,31]. Remarkably, no other study in individual cyclic sports has shown that personal accomplishment was related to changes in the training load of swimmers and rowers [30,31,32]. Additionally, no relationship was found between personal accomplishment and changes in the performance of female cyclists [29], nor a change after a negative life event of runners [33]. This suggests that the feeling of personal accomplishment may not be dependent on increases in physical and psychological stress but on the individual matching of training load between coaches and athletes.

Caution should be taken because we included only 14 speed skaters. Therefore, the chance of a false positive correlation is large. Nevertheless, all the scales which show significant correlations are categorized as recovery scales. This points out that the perceived recovery may be a key factor to explain differences between intended and perceived training load, or that differences in intended–perceived training load interfere with perceived recovery.

## 5. Conclusions

This study found that, during the preparatory phase of 4 weeks, junior speed skaters trained less than was intended by the coach, with no mismatch in RPE and training load calculated for the group. However, we have observed a large variation in the (mis)match, showing a variety between skaters who trained more than intended and skaters who trained less than intended. Additionally, it seems that there are differences between the (mis)match for different training modalities and different training intensities. A (mis)match between the coach and the speed skater is related to changes in recovery scales. That is, when speed skaters approach or exceed the coach’s intended RPE, their perception of success seems to improve. Perception of physical recovery and self-regulation improves when they approach or exceed the intended duration per session, and personal accomplishment increases when they approach or exceed the intended training load. This highlights the importance of looking at each individual’s deviation from the coach and investigating the reasons for deviation in order to find an appropriate intervention to balance the perception of stress and recovery.

## Figures and Tables

**Figure 1 ijerph-19-11221-f001:**
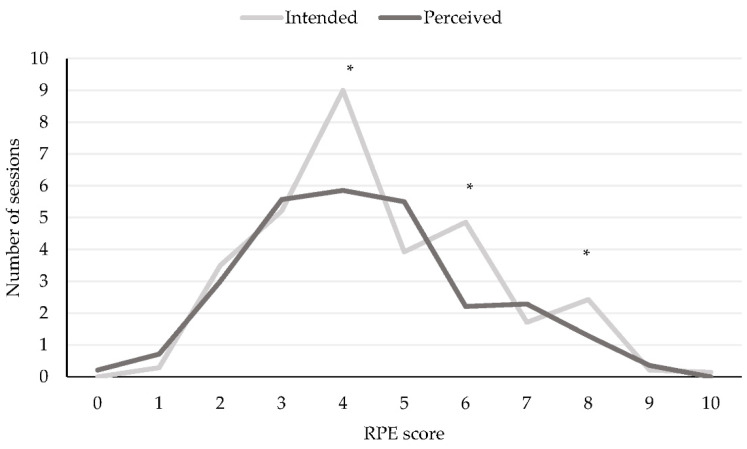
Intended and perceived RPE distribution over all sessions. *: *p* < 0.05.

**Figure 2 ijerph-19-11221-f002:**
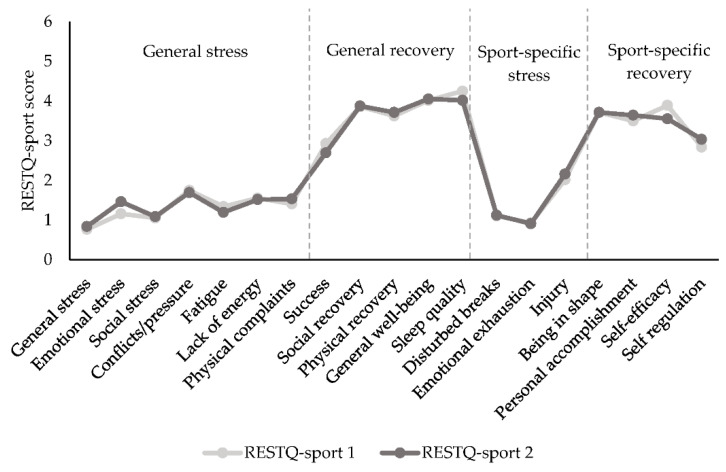
Mean RESTQ-sport scores on the first and second RESTQ-sport measurements. SDs for each subscale range from 0.39 to 1.07. No significant differences were found.

**Table 1 ijerph-19-11221-t001:** Mean values of intended and perceived training variables (n = 14) along with the difference between training variables (perceived minus intended) over four weeks, mean ± SD shown. Total refers to summation over the entire 4-week period.

	Intended	Perceived	(Mis)match Perceived-Intended	(Mis)match Range	(Mis)match Effect Size
Total					
Sessions (number)	31 ± 4	27 ± 6	−4 ± 6 *	−15–6	−0.41
Duration (h:min)	52:37 ± 08:41	45:16 ± 10:09	07:20 ± 09:09 *	−23:05–07:17	−0.40
Load (AU)	13,686 ± 2534	11,609 ± 2898	−2076 ± 2615	−6330–3829	−0.31
Average per session					
RPE	4.6 ± 0.4	4.3 ± 0.4	−0.2 ± 0.4	−0.9–0.6	−0.31
Duration (min)	101 ± 5	101 ± 15	0 ± 14	−10–44	−0.17
Load (AU)	435 ± 31	429 ± 57	−7 ± 49	−84–117	−0.01

Note: * *p* < 0.05. AU = Arbitrary Units.

**Table 2 ijerph-19-11221-t002:** Mean values of intended and perceived training variables for 14 speed skaters along with the difference between training variables (perceived minus intended) for different training modalities over four weeks, mean ± SD shown.

	Intended	Perceived	(Mis)match ∆Perceived–Intended	(Mis)match ∆Range	(Mis)match Effect Size
Speed skating					
Total sessions (number)	5 ± 1	4 ± 1	−1 ± 1	−2–0	−0.28
Total duration (h:min)	06:44	05:41	−01:03 ± 01:20	−03:00–00:20	−0.27
Total load (AU)	2264 ± 550	1733 ± 597	−532 ± 545 *	−1355–340	−0.46
RPE per session	5.7 ± 0.6	5.1 ± 0.5	−0.6 ± 0.7 *	−1.8–0.6	−0.46
Duration per session (min)	79 ± 2	78 ± 5	−1 ± 5	−5–12	−0.15
Load per session (AU)	449 ± 56	396 ± 47	−53 ± 65 *	−147–68	−0.43
Cycling					
Total sessions (number)	14 ± 3	12 ± 4	−2 ± 4	−14–2	−0.21
Total duration (h:min)	27:14	23:34	−03:41 ± 09:39	−29:50–08:02	−0.23
Total load (AU)	6063 ± 1814	5776 ± 2428	−287 ± 2751	−7990–4514	−0.05
RPE per session	3.8 ± 0.6	3.8 ± 0.7	0.0 ± 0.7	−1.8–1.1	−0.02
Duration per session (min)	120 ± 9	120 ± 9	0.0 ± 12	−12–31	−0.14
Load per session (AU)	442 ± 62	475 ± 119	33 ± 128	−211–341	−0.21
Strength training					
Total sessions (number)	7 ± 1	4 ± 2	−3 ± 2 *	−6–−1	−0.78
Total duration (h:min)	10:01	05:48	−04:13 ± 02:06 *	−09:15–−01:30	−0.80
Total load (AU)	2879 ± 593	1602 ± 704	−1276 ± 530 *	−2160–−520	−0.67
RPE per session	4.7 ± 0.7	4.6 ± 0.9	−0.2 ± 0.9	−1.3–1.9	−0.09
Duration per session (min)	89 ± 3	82 ± 9	−6 ± 10 *	−23–12	−0.44
Load per session (AU)	422 ± 57	380 ± 88	−42 ± 83	−146–77	−0.31
Other					
Total sessions (number)	6 ± 2	7 ± 3	1 ± 3	−4–7	−0.24
Total duration (h:min)	08:38	10:14	−01:36 ± 08:00	−06:20–26:35	−0.06
Total load (AU)	2481 ± 658	2499 ± 1519	18 ± 1358	−1880–3350	−0.03
RPE per session	4.8 ± 0.3	4.1 ± 1.4	−0.7 ± 1.4	−5.0–0.7	−0.27
Duration per session (min)	89 ± 6	82 ± 46	−7 ± 48 *	−95–134	−0.46
Load per session (AU)	426 ± 28	348 ± 160	−78 ± 168 *	−470–206	−0.39

Note: * *p* < 0.05. AU = Arbitrary Units.

**Table 3 ijerph-19-11221-t003:** Pearson and Spearman correlations between change scores for the RESTQ-sport (post minus pre) and differences between intended and perceived training variables (perceived minus intended) of 14 speed skaters.

	∆Perceived–Intended			
	Total Sessions	Total Load	RPE Per Session	Duration Per Session ^a^
**∆** General stress ^a^	−0.030	−0.188	0.010	−0.063
**∆** Emotional stress	−0.367	−0.475	−0.205	0.056
**∆** Social stress	0.049	0.014	−0.210	−0.112
**∆** Conflicts/pressure	0.479	0.302	0.090	−0.353
**∆** Fatigue	−0.004	−0.200	−0.321	−0.148
**∆** Lack of energy	−0.010	−0.060	0.249	−0.259
**∆** Physical complaints	−0.247	−0.352	−0.476	0.332
**∆** Success	0.309	0.245	0.568 *	−0.118
**∆** Social recovery	0.363	0.298	0.117	−0.126
**∆** Physical recovery	0.173	0.525	0.389	0.575 *
**∆** General well-being ^a^	0.233	0.370	0.166	−0.009
**∆** Sleep quality	0.038	−0.096	−0.050	−0.186
**∆** Disturbed breaks ^a^	−0.142	0.052	0.139	0.389
**∆** Emotional exhaustion	−0.224	−0.460	−0.275	0.169
**∆** Injury	−0.362	−0.411	−0.155	0.197
**∆** Being in shape	−0.013	0.048	0.079	0.174
**∆** Personal accomplishment	0.391	0.589 *	0.515	0.255
**∆** Self-efficacy	0.131	0.313	0.387	0.065
**∆** Self-regulation	−0.057	−0.025	−0.043	0.598 *
**∆** General stress	−0.053	−0.237	0.024	−0.104
**∆** General recovery ^a^	0.319	0.354	−0.243	0.106
**∆** Sport-specific stress	−0.452	−0.503	0.021	0.357
**∆** Sport-specific recovery	0.182	0.380	0.013	0.416
**∆** Recovery–stress	0.275	0.437	−0.048	0.077

Note: ^a^ Spearman correlation; * *p* < 0.05.

## Data Availability

The data presented in this study are available upon reasonable request to the corresponding author.

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
