# Peer review of "Perceived Training of Junior Speed Skaters versus the Coach’s Intention: Does a Mismatch Relate to Perceived Stress and Recovery?"

_ijerph, 2022, doi:10.3390/ijerph191811221_

Round 1

Reviewer 1 Report

Dera Authors, 

Firstly;  I would like to congratulate for the study, especially for the design and the structure.  I'm only not sure why you use the title with ?  " How do junior speed skaters train versus the coach’s intention and does a mismatch relate to perceived stress and recovery?"  Would be more appropriate to give the reader the ANSWER"...?  I would also sugest to put more focus on teh limitation of the data. 

Best regards, 

Author Response

Dera Authors, 

Firstly;  I would like to congratulate for the study, especially for the design and the structure.  I'm only not sure why you use the title with ?  " How do junior speed skaters train versus the coach’s intention and does a mismatch relate to perceived stress and recovery?"  Would be more appropriate to give the reader the ANSWER"...?  I would also sugest to put more focus on teh limitation of the data. 

Best regards, 

Dear reviewer,

Thank you for the compliments on our paper and for the suggestions. Accordingly, we have made the following changes:

  • The title has been changed into: Perceived training of junior speed skaters versus the coach’s intention. Does a mismatch relate to changes in perceived stress and recovery?
    The new title includes a question because we believe that the answer to the question is too nuanced because of the limitations to be included the title.
  • In order to put more focus on the limitations, we have included the number of athletes in the abstract: line 27.

Reviewer 2 Report

Thank you for the opportunity to review the presentation of your study examining the (mis)match of perceived versus intended training in junior speed skaters. I believe this has merit and will be of interest to the readership - even those who work with athletes other than speed skaters. 

I have a couple of comments and edit suggestions:

1. Do you think the age of the participants affected their perceptions? Could you address RPE, etc. used with this age group briefly in the Introduction and then address if might be a limitation in generalizing to older or elite athletes in the Discussion?

2. Please put numerical results in the abstract.

3. Under Design section 2.2, please change "was" to "were" in the 2nd to last sentence ("....were monitored.").

4. In Training Load section 2.3, perhaps show an equation for the training loads rather than writing out in sentences. Something like: "Intended training load for the coaches was calculated using the following formula: 

Coaches' Training Load = Intended RPE * Intended Duration"    

5. Also, in section 2.3, please put the units for Training Load; I believe they are from your software and are AUs?

6. Inclusion Criteria section 2.5, please change "was" to "were" in the last sentence as data are always plural.

7. Please state what the AU units stand for in Table 1 in the Note section. (Define the abbreviation).

No other comments or edits are suggested by me.

Author Response

Reviewer 2

Thank you for the opportunity to review the presentation of your study examining the (mis)match of perceived versus intended training in junior speed skaters. I believe this has merit and will be of interest to the readership - even those who work with athletes other than speed skaters. 

I have a couple of comments and edit suggestions:

Thank you for reviewing our paper and the suggestions for improvement.

  1. Do you think the age of the participants affected their perceptions? Could you address RPE, etc. used with this age group briefly in the Introduction and then address if might be a limitation in generalizing to older or elite athletes in the Discussion?

Response 1: Thank you for this comment. We briefly added some information on the influence of age on RPE in the introduction in line 90 to 93. Accordingly to your comment, an elaboration on this topic has been added in the discussion in line 309 to 321.

  1. Please put numerical results in the abstract.

Response 2: We have added the numerical results in the abstract.

  1. Under Design section 2.2, please change "was" to "were" in the 2nd to last sentence ("....were monitored.").

Response 3: We have changed the text accordingly.

  1. In Training Load section 2.3, perhaps show an equation for the training loads rather than writing out in sentences. Something like: "Intended training load for the coaches was calculated using the following formula: 

Coaches' Training Load = Intended RPE * Intended Duration"    

Response 4: We have adopted the formulas in the text in section 2.3.

  1. Also, in section 2.3, please put the units for Training Load; I believe they are from your software and are AUs?

Response 5: We have added (AU) to the formulas in the text in section 2.3.

  1. Inclusion Criteria section 2.5, please change "was" to "were" in the last sentence as data are always plural.

Response 6: We have changed the text accordingly.

  1. Please state what the AU units stand for in Table 1 in the Note section. (Define the abbreviation).

Response 7: We have changed the note section accordingly for Table 1 and Table 2.

No other comments or edits are suggested by me.